# HaploVL: A Single-Transformer Baseline for Multi-Modal Understanding

**Rui Yang** [1]  **Lin Song** [2+]  **Yicheng Xiao** [3]  **Runhui Huang** [1]  **Yixiao Ge** [2]  **Ying Shan** [2]  **Hengshuang Zhao** [1*]

## Abstract

Recent advancements in large language models (LLMs) have significantly propelled the development of large multi-modal models (LMMs), highlighting the potential for general and intelligent assistants. However, most LMMs model visual and textual modalities separately, leading to recent efforts to develop native LMMs using a single transformer. Despite the promise, these native models are resource-intensive and often exhibit performance gaps compared to their compositional counterparts. To alleviate this issue, we propose a simple yet efficient method to construct a baseline for the native and end-to-end large multi-modal model in a single transformer. First, we propose a new early-fusion LMM that can fuse multi-modal inputs in the early stage and respond to visual instructions in an auto-regressive manner. Second, we devise an efficient training recipe for the proposed model, which harnesses the prior knowledge of the pre-trained models, addressing both the performance limitations and the challenge of resource consumption. The proposed model demonstrates superior performance compared to other LMMs using one transformer and significantly narrows the performance gap with compositional LMMs. Code is available at https://github.com/Tencent/HaploVLM.

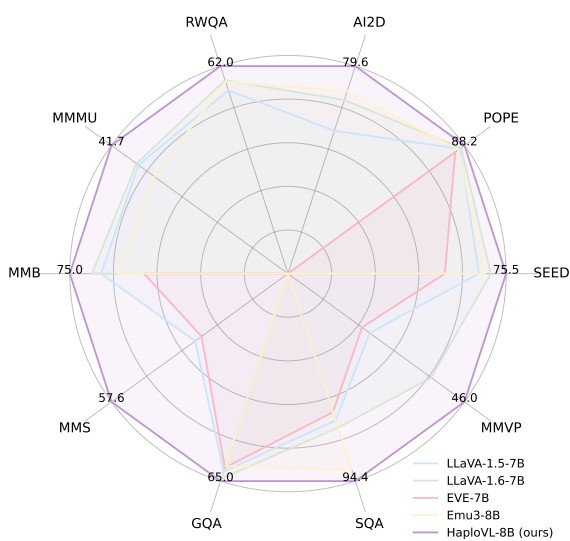

Figure 1: Performance comparison with single-transformer models on multi-modal understanding benchmarks. Our HaploVL demonstrates superiority over other counterparts.

## 1. Introduction

Large language models (LLMs) (OpenAI, 2023; Dubey et al., 2024; Yang et al., 2024a) have recently made significant strides in the realm of artificial intelligence. This progress has substantially accelerated the development of large multi-modal models (LMMs), which include both proprietary commercial models (Achiam et al., 2023; Team et al., 2024) and open-source models (Dai et al., 2023; Zhu

et al., 2023a; Liu et al., 2024a). These models facilitate complex vision-language dialogues and interactions. The majority of open-source models (Liu et al., 2024c; Dai et al., 2023) leverage one or more separate vision components to model the visual modality, thus equipping LLMs with visual understanding and reasoning capabilities. For instance, the LLaVA series (Liu et al., 2024c;a) directly harnesses the pretrained CLIP vision encoder (Radford et al., 2021) to extract high-level vision embeddings and uses a projector to connect these embeddings with LLMs.

Since language is human-generated signals that have already been abstracted (He et al., 2022), the text embeddings produced by the word embedding layer contain semantic information and are high-level. Therefore, it is reasonable to combine text embeddings with vision embeddings from a pre-trained vision encoder, as both types of embeddings are semantic. However, the off-the-shelf vision encoder (Radford et al., 2021) tends to produce highly compressed global semantics and neglect fine-grained visual information. Thus, it may fail to extract effective visual cues required by the text, leading to difficulties for LMMs in handling fine-grained tasks (Tong et al., 2024).

[1]The University of Hong Kong [2]ARC Lab, Tencent PCG [3]Tsinghua University. Team leader: Lin Song <ronnysong@tencent.com>. Correspondence to: Hengshuang Zhao <hszhao@cs.hku.hk>.

*Proceedings of the 42nd International Conference on Machine Learning*, Vancouver, Canada. PMLR 267, 2025. Copyright 2025 by the author(s).

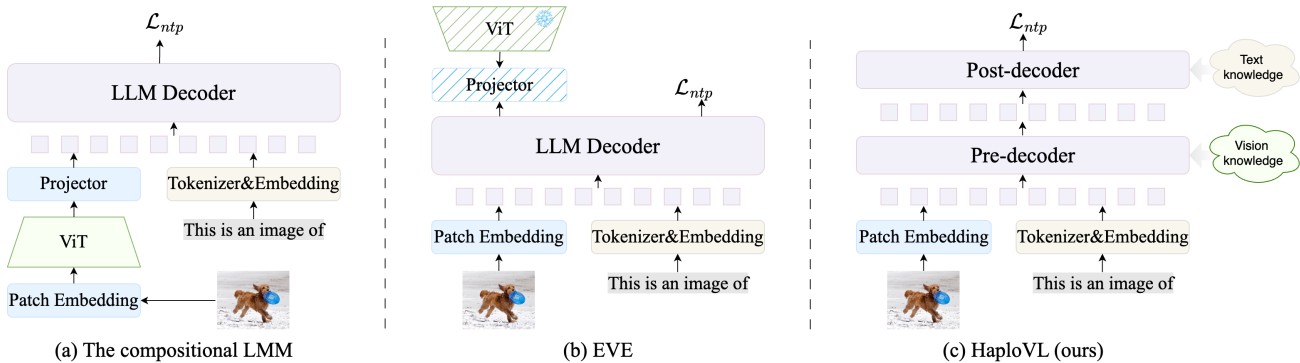

(a) The compositional LMM                     (b) EVE                     (c) HaploVL (ours)

Figure 2: Architecture comparison with the compositional LMM (Liu et al., 2024a), EVE (Diao et al., 2024). In our HaploVL, the pre-decoder dynamically extracts vision cues based on the input text, and the post-decoder further fuses the multi-modal embeddings. Our model inherits the prior knowledge from vision and language models, thus requiring less data than EVE.

| Method | Data (M) | Early fusion | Seamntic align | SEED | MMStar$_{FP}$ | MMVP |
|--------|----------|--------------|----------------|------|---------------|------|
| LLaVA-1.5-7B | 1.2 | ✗ | ✔ | 66.1 | 24.0 | 21.3 |
| EVE-7B | 35.0 | ✔ | ✗ | 54.3 | 24.2 | 19.3 |
| HaploVL-7B | 1.2 | ✔ | ✔ | 67.5 | 28.9 | 24.7 |

Table 1: Comparison with compositional LLaVA (Liu et al., 2024a) and unified EVE (Diao et al., 2024) on multimodal benchmarks: SEED-Bench (Li et al., 2023a), the fine-grained split of MMStar (Chen et al., 2024a), and MMVP (Tong et al., 2024).

To address this issue, we propose an early-fusion LMM named HaploVL. Our model fuses the vision and text embeddings at an early stage, enabling text embeddings to autonomously acquire the necessary vision cues. Specifically, HaploVL uses a lightweight patch embedding layer, *a single linear layer*, to embed visual input and a text embedding layer to process textual inputs. Subsequently, the transformer backbone extracts the necessary vision information based on the text input and generates language responses according to the resulting fused representations.

Some recent studies (Bavishi et al., 2023; Team, 2024; Diao et al., 2024) also fall under the category of early-fusion LMMs and have endeavored to develop a unified multimodal transformer with a concise inference process. For example, Fuyu (Bavishi et al., 2023) directly utilizes a simple linear layer instead of a vision encoder to embed the input image and leaves the mixed modality sequence to the subsequent transformer. EVE (Diao et al., 2024) aims to replicate Fuyu by being distilled from a fixed vision encoder, thus reducing the training data. However, it forces alignment between a large language model (7B) and a small ViT (300M) without allowing the LMM to learn from high-level vision features. Therefore, there is a significant performance gap between it and compositional LMMs on vision-language benchmarks, despite using 35M training data.

To this end, our HaploVL utilizes a pre-decoder to autonomously acquire the necessary vision cues according to text information, and a post-decoder to further process the extracted high-level multi-modal embeddings. Since training such a model from scratch is very expensive, *e.g.,* the energy consumption required to pre-train the Chameleon-30B (Team, 2024) is equivalent to what is needed to power a Tesla Model 3 to travel around the equator for about 225 times [1], we propose to leverage prior knowledge acquired from pre-trained models. This is because the pre-trained models have gained extensive knowledge by training on massive data, *e.g.,* the CLIP vision encoder (Radford et al., 2021) obtained vision-based knowledge by seeing billions of images, and Llama (Dubey et al., 2024) gained text-based knowledge by seeing trillions of text tokens. Specifically, the pre-decoder inherits prior vision knowledge from a vision encoder while simultaneously processing text and vision modalities to perform modal expansion. Plus, the LLM retains its prior text knowledge and learns to take vision embeddings as a condition. In this way, we significantly reduce the required data and training costs in comparison to other early-fusion and single-transformer LMMs (Bavishi et al., 2023; Team, 2024; Diao et al., 2024; Wang et al., 2024b), and bridge the performance gap between unified and compositional LMMs. As shown in Table 1, HaploVL achieves a significant performance improvement over LLaVA and EVE (Diao et al., 2024) on fine-grained perception benchmarks (Chen et al., 2024a; Tong et al., 2024). This demonstrates promising potential for developing multi-modal models with a single transformer efficiently.

Our contributions can be summarized as follows:

- We develop a new early-fusion LMM with a single transformer that acquires the necessary vision cues

---

[1]Estimated by Chameleon-30B (Team, 2024)'s GPU hours, A100 GPU's power, and Tesla Model 3's power consumption.

in the early stage and generates language responses conditioned on fused multi-modal embeddings.

- We design an efficient training recipe for the proposed model, which leverages the prior knowledge from pre-trained models. This approach not only reduces the need for large-scale data and computational resources but also bridges the performance gap between the unified and compositional LMMs.

## 2. Related Work

**Encoder-decoder large multi-modal models** as exemplified by LLaVA (Liu et al., 2024c), employ a pre-trained vision encoder like CLIP (Radford et al., 2021) to extract visual embeddings and an MLP layer to align the visual embeddings with large language models (LLMs). Then, these models with the "Encoder-MLP-LLM" configuration are fine-tuned on tailored instruction data to obtain the capability of image understanding and reasoning. Numerous innovations have sought to improve the performance of this method by utilizing more powerful vision encoders such as SigLIP (Zhai et al., 2023) and InternViT-6B (Chen et al., 2024b), expanding the input size to any resolution (Liu et al., 2024a), and synthesizing high-quality data (Li et al., 2024a). At the same time, inspired by this straightforward architecture, numerous studies have replaced the vision encoder with a domain-specific encoder to develop a modality-specific multi-modal model (Chu et al., 2023; Qi et al., 2024). Plus, others (Lu et al., 2022a; Zhan et al., 2024) integrate multiple modality-specific encoders with the language model to enable it to accommodate more additional modalities. However, a significant limitation of this method is the lengthy visual sequences. To alleviate this issue, BLIP-2 (Li et al., 2023b) develops a Q-former to replace the long visual features with a fixed number of learnable queries. This "Encoder-Q-former-LLM" configuration has been replicated by many studies (Zhu et al., 2023a; Dai et al., 2023; Ye et al., 2023; Li et al., 2024c). MiniGPT-4 (Zhu et al., 2023a), InstructBLIP (Dai et al., 2023), mPLUG-Owl (Ye et al., 2023), and MGM (Li et al., 2024c). Apart from those architectures, Flamingo (Alayrac et al., 2022) incorporates gated cross-attention layers into LLMs for cross-modality alignment; CogVLM (Wang et al., 2023) equips extra vision experts in each block of LLMs to enable visual-language feature alignment; and mPLUG-Owl-2 (Ye et al., 2024) introduces a modality-adaptive module to decouple vision-language representations.

**Single-transformer multi-modal models** aim to discard the vision encoder and merely allow the language model to process text embeddings and vision embeddings that are not fully compressed. Fuyu (Bavishi et al., 2023) utilizes a linear projector to patchify the raw image wherein the obtained low-level vision patch embeddings are treated as contiguous tokens. Compared with models with the "Encoder-MLP-LLM" configuration, Fuyu directly fuses low-level vision embeddings with text embeddings instead of high-level vision embeddings (hidden states of the vision encoder). Besides, Chameleon (Team, 2024) employs a VQ codebook (Van Den Oord et al., 2017) to discretize the image to a set of discrete visual tokens, akin to the process of the text tokenizer. Thus, the vision and text embedding can be extracted from the same embedding layer and processed by a decoder-only transformer. Emu3 (Wang et al., 2024b) has extended this streamlined pipeline to generate high-quality images and videos. Since these methods are trained from scratch, they consume substantial computing resources and necessitate significant amounts of data. To adapt an off-the-shelf decoder-only language model to a multi-modal model, EVE (Diao et al., 2024) introduces a meticulously designed patch embedding layer and training strategies. However, they still exhibit a significant performance gap compared to encoder-decoder multi-modal language models, despite utilizing 35 M images.

## 3. Method

Our HaploVL is a single-transformer multi-modal model. Like popular LMMs (Liu et al., 2024c; Dai et al., 2023; Bavishi et al., 2023), it maps visual and textual input to the same latent space and takes them as conditions for text generation in an auto-regressive manner. Unlike other LMMs that always rely on highly compressed vision embeddings from a fixed vision encoder, our HaploVL fused the visual and textual input in the early stage and extracts the necessary vision information based on the text input. Compared to previous early-fusion and single-transformer LMMs (Bavishi et al., 2023; Diao et al., 2024; Team, 2024), our HaploVL is more efficient in training, as it absorbs the prior knowledge that the model learned. In the subsequent section, we begin by presenting a detailed description of HaploVL's architecture, followed by a receipt of the efficient training procedure.

### 3.1. Architecture

From a holistic perspective, as illustrated on the right side of Figure 3, our HaploVL adopts a multi-modal end-to-end transformer architecture. Most of its parameters are attributed to the transformer decoder that processes sequences, regardless of the modality. HaploVL can generate the language response $X_a$ conditioned on the visual input $X_v$ and textual input $X_t$ in an auto-regressive manner. This generation process is as clear as that of language models (Radford et al., 2019; Dubey et al., 2024). Given a sequence of length $L$, HaploVL calculates the probability of $X_a$ by:

$$p(X_a|X_v, X_t) = \prod_{i=1}^{L} p_\theta(x_i|X_v, X_t, X_{a,<i}), \quad (1)$$

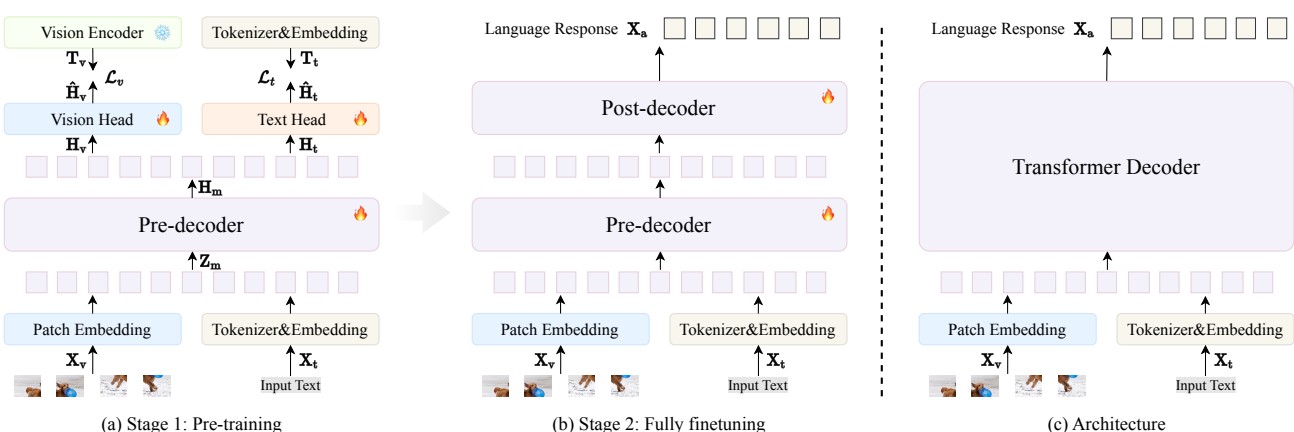

(a) Stage 1: Pre-training    (b) Stage 2: Fully finetuning    (c) Architecture

Figure 3: The diagram of HaploVL. It includes a transformer decoder made up of a pre-decoder and a post-decoder. During the pre-training stage (a), the pre-decoder is trained by distilling knowledge from the pre-trained vision encoder and the text embeddings of the LLM. Heads and teacher models are dropped after pre-training. In the full fine-tuning stage (b), the entire model is fine-tuned using visual instruction data.

where $X_{a,<i}$ denotes the answer tokens before the current prediction token $x_i$. $\theta$ is the parameter of components that model conditional probability. Thus, $\theta$ of our HaploVL is from the whole model, while $\theta$ of compositional LMMs using separated vision encoders (Liu et al., 2024c; Karamcheti et al., 2024) is the parameter of the LLM.

From a detailed perspective, as depicted in Figure 3 (b), HaploVL can be decomposed into three primary components: (1) multi-modal embedding layers, (2) a pre-decoder, and (3) a post-decoder. These bottom-up modules work together to facilitate efficient training and enhance visual understanding and reasoning performance, especially in the fine-grained scene.

**Multi-modal embedding layers.** Regarding the input data, we use lightweight and modality-specific components with unshared parameters to map them into a shared latent space $\mathbb{R}^d$. Specifically, for the input RGB image $X_v$, we apply a simple patch embedding layer, *a single linear layer*, to compress local windows ($k \times k$) of pixels into a vision embedding $Z_v$ within the shared latent space $\mathbb{R}^d$. This approach differs from existing compositional LMMs (Liu et al., 2024a; Karamcheti et al., 2024), which typically rely on one or more separate vision encoders to embed visual input. For the input text $X_t$, we leverage the pre-trained LLM's embedding matrix $\mathcal{W}$ to convert each text token into a vector within LLM's space $\mathbb{R}^l$. These text vectors are then projected into text embeddings $Z_t$ within the shared latent space $\mathbb{R}^d$ by a text projector, also *a single linear layer*. The resulting vision and text embeddings, $Z_v$ and $Z_t$, are combined to form a mixed multi-modality embedding sequence $Z_m$, which is fed into the subsequent transformer.

**Pre-decoder.** Upon the multi-modal sequence $Z_m$, the pre-decoder fuses it in the initial stage of HaploVL, extract-

ing visual cues based on text embeddings. Then, it yields a multi-modal hidden state $H_m$. Each block of the pre-decoder consists of a multi-head self-attention layer and a 2-layer MLP with GELU (Hendrycks & Gimpel, 2016) nonlinearity in between. Its configuration, such as depth and width, mirrors that of the vision transformer, as it is required to inherit the prior vision knowledge of a pre-trained vision model. In practice, we default to leveraging CLIP-ViT-L (Radford et al., 2021) which has 24 blocks and an embedding dimension of 1024. Notably, although the pre-decoder inherits prior knowledge from a vision encoder, it differs from the vision encoder. For one thing, the pre-decoder can process both visual and textual input, whereas the vision encoder only processes visual input. Furthermore, the text embeddings in the pre-decoder utilize a causal mask strategy, allowing the pre-decoder to predict the next token in an auto-regressive manner.

**Post-decoder.** Based on the multi-modal hidden state $H_m$, the post-decoder further processes it and outputs a language response. Each block of the post-decoder mirrors the Llama block (Dubey et al., 2024) as it needs to acquire prior textual knowledge the Llama model. Leveraging the inherited knowledge from extensive text data, the post-decoder can swiftly learn multi-modal knowledge and generate language responses based on multi-modal hidden states.

**Masking strategy.** HaploVL employs a mixed masking strategy within its self-attention layers. In a mixed multi-modal sequence, a causal mask is utilized for the textual part. This is consistent with GPT-like language models (Radford et al., 2019). For the visual part, a bidirectional mask is applied to embeddings from a single image, as correlations exist between image tokens regardless of their positional order. In addition, a causal mask is employed between

multiple images, reflecting the temporal causal relationships in sequential data. This modeling approach aligns with prevalent vision models (Dosovitskiy et al., 2021).

## 3.2. Training

We use a two-stage training recipe for HaploVL, as illustrated in Figure 3. In the first stage, the pre-decoder is trained through feature distillation. This enables it to effectively process both visual and textual inputs simultaneously, laying the foundation for subsequent stages. In the second stage, the model is trained to follow visual instruction, which equals LLaVA's visual instruction tuning.

**Stage 1: Pre-training.** As mentioned before, the pre-decoder inherits prior vision knowledge from the pre-trained ViT and can fuse visual and textual inputs. This stage mainly endows the pre-decoder to support vision and text modality. As illustrated in Figure 3 (a), the knowledge distillation approach (Hinton, 2015) is employed to train the pre-decoder, prompting the model to learn new text knowledge and avoiding the model forgetting the inherited vision knowledge. Given the visual input $X_v$ and textual input $X_t$, the output of the pre-decoder is the hidden state $H_m$, which can be decomposed into the visual hidden state $H_v$ and textual hidden state $H_t$ based on their respective token positions.

To preserve the image processing capabilities of the pre-decoder, we adopt the pre-trained CLIP vision encoder (Radford et al., 2021) as a teacher model to guide the expansion process. This approach enables the pre-decoder to retain its inherited knowledge, ensuring its image abilities are not compromised. This vision loss can be formulated as:

$$\mathcal{L}_v = 1 - \frac{1}{hw} \sum_{i=1}^{hw} \cos(\hat{H}_{v,i}; T_{v,i}), \qquad (2)$$

where $\hat{H}_v$ is the projected $H_v$ by a vision head; $T_v$ is the feature extracted from CLIP vision encoder; and $hw$ represents the number of vision embedding after patch partition.

For the textual input, the pre-decoder performs a simple identity mapping. This training target enables the pre-decoder to leverage the strengths of the post-decoder in handling complex generation tasks, thus alleviating the challenging multi-modal learning. More importantly, language is semantic (He et al., 2022). When the text and image are jointly input into the pre-decoder in a mixed way, semantic text embeddings can autonomously acquire the necessary vision cues from raw vision embeddings. For the split textual hidden state $H_t$, we utilize a learnable text head to align it with the teacher embedding, resulting in $\hat{H}_t$. We employ two types of loss functions to encourage this distillation.

(a) The first type is feature loss, which is formulated as:

$$\mathcal{L}_{feat} = 1 + \frac{1}{S} \sum_{i=1}^{S} \left[ \left\| \hat{H}_{t,i} - T_{t,i} \right\|_2 - \cos(\hat{H}_{t,i}; T_{t,i}) \right]. \quad (3)$$

Here, $S$ denotes the length of text tokens in the input sequence; and $T_t$ represents the text embedding directly obtained from the embedding matrix $\mathcal{W}$ using indices of text tokens, which also serves as the input of the pre-decoder. Equation (3) involves the $L_2$ distance to align the magnitude and the cosine loss function to align the direction. This is because the magnitude and direction of text embeddings are crucial for alignment with the post-decoder, which has been retained in a certain input mode when inheriting knowledge from a pre-trained LLM.

(b) The second type of loss function is a current token prediction loss which can be formulated as:

$$\mathcal{L}_{ctp} = -\frac{1}{S} \sum_{i=1}^{S} \sum_{c=1}^{C} y_{i,c} log \left( \frac{e^{\frac{x_{i,c}}{\tau}}}{\sum_{j=1}^{C} e^{\frac{x_{i,j}}{\tau}}} \right). \qquad (4)$$

In this loss function, $C$ refers to the vocabulary size of the tokenizer; $y_i$ is the one-hot label of the $i$-th token; and $y_{i,c}$ is the label of $c$-th word in the vocabulary. $x_i = \hat{H}_{t,i} \cdot \mathcal{W}^T$ is the logit of $\hat{H}_{t,i}$. A learnable temperature $\tau$ is used to adjust the distribution of the logits as CLIP (Radford et al., 2021). This is very effective in reducing the magnitude of the output text embeddings, as the cross-entropy loss function minimizes the total loss by enlarging the logit magnitude. The difference between Equation (4) and the next token prediction loss (Radford et al., 2019) lies in that the target in Equation (4) is derived from the current token instead of the next token.

So far, we have introduced two types of loss functions, and the total text loss function used in the modal expansion stage is the sum of them: $\mathcal{L}_t = \mathcal{L}_{feat} + \mathcal{L}_{ctp}$. We combine interleaved image-text data and pure text data to train the pre-decoder. After modal expansion, we keep the pre-decoder, while discarding the heads.

**Stage 2: Fully fine-tuning.** This training stage is mainly for multi-modal learning. As illustrated in Figure 3 (b), we fine-tune all components of HaploVL in this stage. The next token prediction loss (Radford et al., 2019) is still adopted to maximize the log-likelihood of Equation (1). After tuning, our HaploVL performs capabilities in following human visual instructions.

## 4. Experiment

In this section, we first outline the experimental setup including training settings and dataset. Then, we compare our HaploVL with leading methods on various benchmarks. Finally, an analysis of training procedures and some qualitative results are given at the end of this section.

| Method | Base LLM | SEED | POPE | AI2D | RWQA | MMMU | MMB | MMS | VQAv2 | GQA | SQA | MMVP |
|---|---|---|---|---|---|---|---|---|---|---|---|---|
| *Compositional LMM* | | | | | | | | | | | | |
| InstructBLIP (Dai et al., 2023) | Vicuna-7B | 58.8 | - | 33.8 | 37.4 | 30.6 | 36.0 | - | - | 49.2 | 60.5 | 16.7 |
| LLaVA-1.5 (Liu et al., 2024a) | Vicuna-7B | 66.1 | 85.9 | 54.8 | 54.8 | 35.3 | 64.3 | 30.3 | 78.5* | 62.0* | 66.8 | 21.3 |
| LLaVA-1.6 (Liu et al., 2024b) | Vicuna-7B | 70.2 | 86.5 | 66.6* | 57.8 | 35.8 | 67.4 | - | 81.8* | 64.2* | 70.1 | 37.3 |
| ShareGPT4V (Chen et al., 2023) | Vicuna-7B | - | - | 58.0 | 54.9 | 37.2 | 68.8 | 33.0 | 80.6* | 63.3* | 68.4 | - |
| VILA (Lin et al., 2024) | Llama-2-7B | 61.1 | 85.5 | - | - | - | 68.9 | - | 80.8* | 63.3* | 73.7 | - |
| LLaVA-OV (Li et al., 2024a) | Qwen2-7B | 75.4 | - | 81.4* | 66.3 | 48.8 | 80.8 | 61.7 | - | - | 96.0* | - |
| *Single-Transformer LMM* | | | | | | | | | | | | |
| Fuyu-8B (Bavishi et al., 2023) | Persimmon-8B | - | 74.1 | 64.5 | - | 27.9 | 10.7 | - | 74.2 | - | - | - |
| Chameleon-30B (Team, 2024) | - | - | - | - | - | - | 37.6 | - | 69.6 | - | - | - |
| EVE-7B (Diao et al., 2024) | Vicuna-7B | 54.3 | 83.6 | - | - | - | 49.5 | 28.2 | 75.4* | 60.8* | 63.0 | 19.3 |
| Emu3-8B (Wang et al., 2024b) | - | 68.2 | 85.2 | 70.0* | 57.4 | 31.6 | 58.5 | - | 75.1* | 60.3* | 89.2* | - |
| HaploVL-8B (ours) | Llama-3-8B | 75.1 | 88.6 | 79.2* | 61.4 | 37.4 | 73.6 | 57.2 | 81.0* | 65.5* | 95.3* | 45.3 |
| HaploVL-8B-MI (ours) | Llama-3-8B | 75.5 | 88.2 | 79.6* | 62.0 | 41.7 | 75.0 | 57.6 | 80.7* | 65.0* | 94.4* | 46.0 |
| HaploVL-7B-Pro (ours) | Qwen2.5-7B | 75.0 | 88.7 | 80.6* | 64.3 | 48.7 | 80.5 | 61.4 | 81.1* | 64.6* | 96.9* | 50.1 |

Table 2: Comparison on multi-modal benchmarks, including SEED (Li et al., 2023a), POPE (Li et al., 2023c), AI2D (Kembhavi et al., 2016), RWQA (x.ai, 2024), MMMU (Yue et al., 2024), MMB (Liu et al., 2024d), MMStar (Chen et al., 2024a), VQAv2 (Goyal et al., 2017), GQA (Hudson & Manning, 2019), and SQA (Lu et al., 2022b). '*' denotes images of related training datasets are observed during training. HaploVL-8B-MI is the model further fine-tuned on multi-image datasets.

## 4.1. Experiment Setup

**Implementation details.** In this study, we instantiate HaploVL by allowing a pre-decoder to receive images and texts at the same time. The pre-decoder inherited the vision knowledge of the CLIP-ViT-L (Radford et al., 2021). The post-decoder inherits the text knowledge from Vicuna-7B (Chiang et al., 2023) and Llama-3-8B (Dubey et al., 2024), resulting in HaploVL-7B and HaploVL-8B, respectively. During the pre-training stage, we optimize the post-decoder for 40 K steps with $1 \times e^{-4}$ learning rate, a batch size of 256, and 2 K warm-up steps. In terms of the data, all models are trained on 665 K plus 558 K multi-modal samples from LLaVA-1.5 (Liu et al., 2024a) if there is no other statement. During the fully fine-tuning stage, the learning rate is set to $2 \times e^{-5}$ and batch size to 128. Regarding the data, our best model is optimized on the 4 M visual instruction data for 1 epoch ( 30K steps). For HaploVL-7B, we align it with LLaVA (Liu et al., 2024c). Thus, we first tune the connector between the pre-decoder and post-decoder using 558 K caption data and then fully tune the model using 665 K instruction data. For HaploVL-8B with the ability to input any resolution, we first tune the whole model using 1.2 M caption data (Chen et al., 2023) and then tune the model using 4 M instruction data (Li et al., 2024a). For the models that support the multi-image and video input, we continue training the single-image model using the mix of interleaved data and single-image data. For the ablation experiments, the models are optimized on the 0.6 M visual instruction data for 5 K steps. All models are optimized using the AdamW (Loshchilov & Hutter, 2019) optimizer and cosine scheduler on 32 GPUs with 64GB per-device

memory. More details are recorded in the Appendix. The data is mainly from LLaVA (Liu et al., 2024a; Li et al., 2024a), dolphin (Computations, 2023), CC3M (Changpinyo et al., 2021), and COCO (Lin et al., 2014).

**Pre-training data.** The main samples include 665 K visual instruction data (Liu et al., 2024c) and 665 K textual instruction data from dophin (471 K) (Computations, 2023), Alpaca (51 K) (Taori et al., 2023), and ShareGPT (143 K) (ShareGPT, 2023). The auxiliary samples involve 558 K image caption data (Liu et al., 2024c) and 1 M pure text data (Computations, 2023). The auxiliary samples are randomly combined with the main samples during training. Assume a main sample is $S_m$ and a auxiliary sample is $S_a$, the combined samples can be one of $< S_a, S_m >$ and $< S_m, S_a >$. In this way, we obtain interleaved samples.

**Single-image instruction data.** 665 K instruction data is from LLaVA-1.5 (Liu et al., 2024a). 4 M instruction data is partly from LLaVA-OneVision (Li et al., 2024a), where we filter some error samples.

**Multi-image instruction data.** We use the interleaved data collected by (Li et al., 2024b) to endow the model (HaploVL-8B-MI) with the ability to process multiple images.

**Evaluation data.** Moreover, HaploVL is evaluated on widely adopted image-based benchmarks including GQA (Hudson & Manning, 2019), VQAv2 (Goyal et al., 2017), ScienceQA-IMG (SQA) (Lu et al., 2022b), AI2D (Kembhavi et al., 2016), MMBench-EN-dev (MMB) (Liu et al., 2024d), MMMU (Yue et al., 2024), Real-WorldQA (x.ai, 2024), MMStar (MMS) (Chen et al., 2024a),

POPE (Li et al., 2023c), SEED-Bench-IMG (SEED) (Li et al., 2023a), and MMVP (Tong et al., 2024). Among these benchmarks, MMVP mainly focuses on fine-grained perception.

## 4.2. Main Results

We compare our model with existing multi-modal models, including both separate models and unified models with a single transformer, in Table 2. Notably, our model achieves superior performance compared to other unified models. Specifically, we outperform Emu3 (Wang et al., 2024b) by 15.1% on the MMBench (Liu et al., 2024d) and by 5.5% on the MMMU (Yue et al., 2024). Additionally, our model significantly surpasses EVE (Diao et al., 2024), a model that uses pre-trained weights, with a lead of 24.1% on MM-Bench (Liu et al., 2024d) and 20.8% on SEED-Bench (Li et al., 2023a). These results demonstrate the promising potential of our model in multi-modal capabilities. Furthermore, we also compare our model with separate models and find that our model has a significant advantage over previous separate models (Dai et al., 2023; Liu et al., 2024a; Chen et al., 2023; Lin et al., 2024). However, our performance still falls short of the state-of-the-art separate open-source models LLaVA-OneVision (Li et al., 2024a). We attribute this to the input resolution and context length. LLaVA-OneVision (Li et al., 2024a) uses 7290 tokens to represent an input image, while our model only uses up to 2304 tokens. Due to computational resource constraints, we can only set the context length to 6144, which affects the model's effectiveness to some extent. Nevertheless, the performance of HaploVL-7B-Pro is nearly comparable to that of LLaVA-OneVision. Plus, we achieve a simple and efficient baseline for one multi-modal transformer, which outperforms other native LMMs using fewer resources. We expect to further improve the performance of such models based on this foundation.

## 4.3. Ablation Study

**Ablation for different LLMs, resolution, and visual instruction data.** As shown in Table 3, we achieve enhanced performance by upgrading the language model, input resolution, and instruction data. Specifically, employing a more advanced language model (Llama-3 (Dubey et al., 2024)) yields an average performance gain of 2.5%. This highlights that multi-modal understanding capabilities are correlated with the capabilities of the language model.

Increasing the resolution from $336 \times 336$ to $672 \times 672$ results in an average performance improvement of 3.3% using the same 665K dataset, especially showing a notable 3.7% gain on POPE (Li et al., 2023c). This underscores the importance of enabling the LMM to perceive finer-grained visual details. When expanding the visual instruction data at the $672 \times 672$

| Base LLM | Res. | VID | Avg | GQA | POPE | MMS | MMVP |
|---|---|---|---|---|---|---|---|
| Vicnua-7B | 336 | 665K | 51.8 | 62.5 | 85.4 | 34.5 | 24.7 |
| Llama-3-8B | 336 | 665K | 54.3 | 63.1 | 84.8 | 39.4 | 30.0 |
| Llama-3-8B | 672 | 665K | 57.6 | 65.7 | 88.2 | 42.0 | 34.3 |
| Llama-3-8B | 672 | 4M | 64.2 | 65.5 | 88.6 | 57.2 | 45.3 |

Table 3: Ablation for different LLMs, resolution (Res.), and visual instruction data (VID).

| Method | ST | MMVP | MMS | | | | | | |
|---|---|---|---|---|---|---|---|---|---|
| | | | Avg | CP | FP | IR | LR | ST | MA |
| LLaVA-1.5-7B | ✗ | 21.3 | 30.3 | 58.8 | 24.0 | 38.8 | 24.0 | 13.6 | 22.8 |
| EVE-7B | ✔ | 19.3 | 28.2 | 40.6 | 24.2 | 32.7 | 27.5 | 20.4 | 23.8 |
| HaploVL-7B | ✔ | 24.7 | 34.5 | 63.8 | 28.9 | 38.5 | 33.6 | 17.6 | 24.6 |

Table 4: In-depth comparison with LLaVA-1.5-7B (Liu et al., 2024a) and EVE (Diao et al., 2024) on MMVP (Tong et al., 2024) and MMStar (Chen et al., 2024a). 'ST' denotes whether the model belongs to the single-transformer LMM. CP: coarse perception, FP: fine-grained perception, IR: instance reasoning, LR: logical reasoning, ST: science and technology, and MA: mathematics.

resolution, the average performance improves by 6.6% since LMM's knowledge is enriched. These gains are particularly pronounced on benchmarks such as MMStar (Chen et al., 2024a) and MMVP (Tong et al., 2024), suggesting that fine-grained perception ability can be enhanced after expanding LMM's vision knowledge. However, a slight performance decline is observed on GQA (Hudson & Manning, 2019). This discrepancy may stem from differences in the distribution of the 4M instruction data compared to the GQA dataset.

**Compared with the compositional LMM using the same LLM and training data.** We compare the performance of our method with that of LLaVA-1.5-7B (Liu et al., 2024a), a typical compositional LMM, using the same LLM (Vicuna-7B) and instruction data (665K) from (Liu et al., 2024a). Since the LLM and instruction data primarily affect performance on different benchmarks, we restrict the data to ensure a fair comparison between our method and LLaVA-1.5-7B. This allows us to verify whether the LMM using one single transformer has advantages over separate models. As shown in Table 4, on the MMVP (Tong et al., 2024) benchmark, our model obtains 3.4% and 5.4% gains than LLaVA-1.5-7B and EVE-7B (Diao et al., 2024), respectively; and, on the MMStar (Chen et al., 2024a) benchmark, our model outperforms LLaVA-1.5-7B and EVE-7B by 4.2% and 6.3%, respectively. We further analyze the detailed scores on MM-Star (Chen et al., 2024a), including coarse perception (CP), fine-grained perception (FP), instance reasoning (IR), logical reasoning (LR), science and technology (ST), and mathematics (MA). Notably, our HaploVL-7B model exhibits a

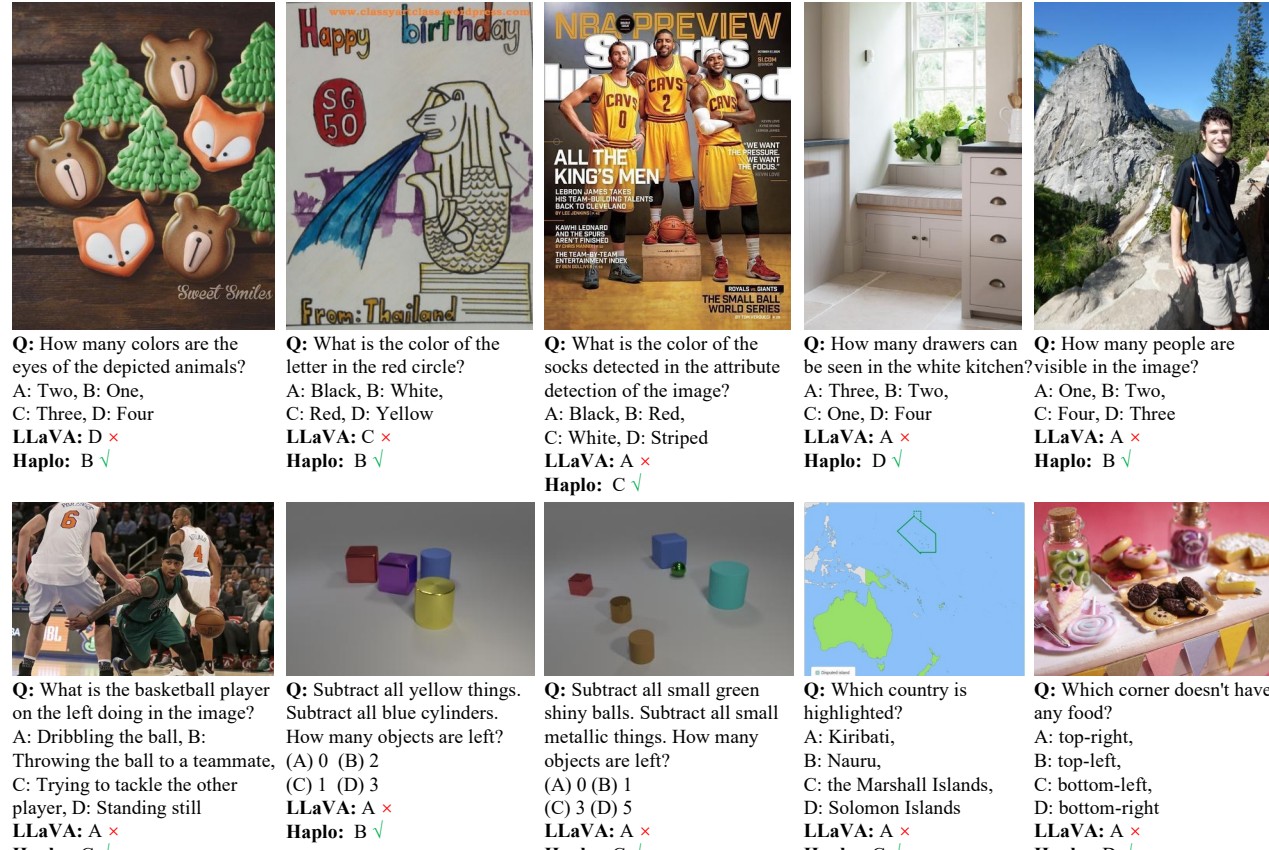

Figure 4: Qualitative comparison of LLaVA-1.5-7B (Liu et al., 2024a) and our HaploVL-7B. The first line involves cases about fine-grained perception. The second line includes cases of logical reasoning that depend on fine-grained perception.

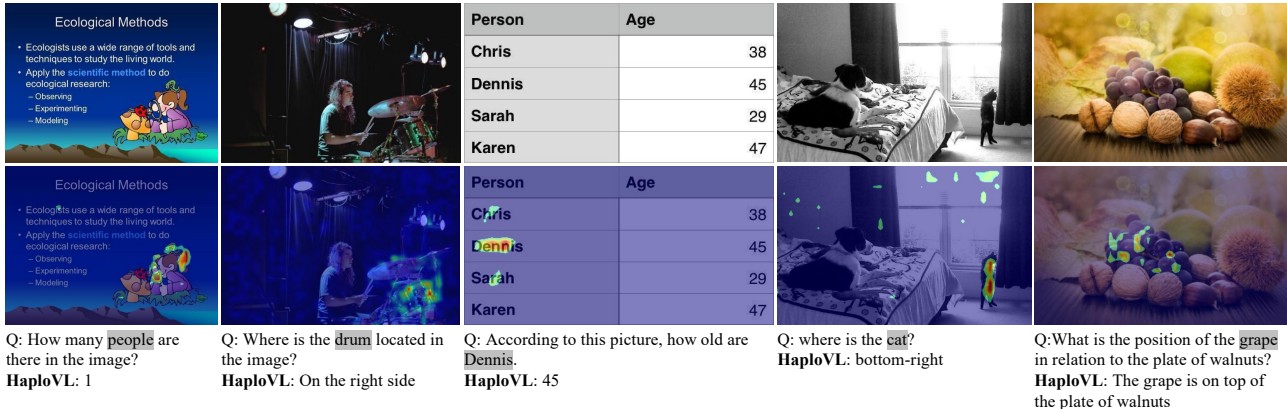

Figure 5: Visualization for the early fusion mechanism of our single transformer. The second row illustrates the attention map of the gray words concerning the vision embeddings after the pre-decoder.

4.9% improvement in fine-grained perception and a 9.6% improvement in logical reasoning over LLaVA-1.5-7B. This suggests that fusing raw image and text embeddings in a single transformer is beneficial for fine-grained perception and subsequently enhances image-based logical reasoning. In contrast, separate models using high-level semantic embeddings from CLIP-ViT encoder (Radford et al., 2021) directly may obscure fine-grained image information, thereby impairing the model to perform tasks that rely on image details. This is consistent with previous study (Tong et al., 2024).

To further illustrate the differences in fine-grained perception and logical reasoning, we provide qualitative results in Figure 4. The first row shows cases of fine-grained per-

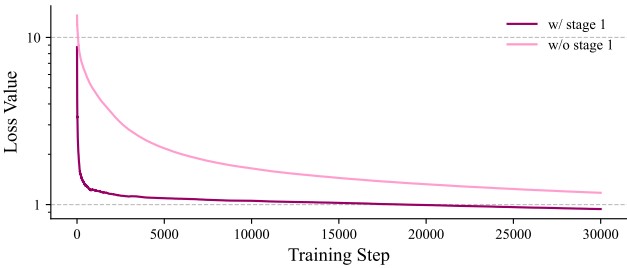

Figure 6: Loss curve of the full fine-tuning stage. We use ema to smooth the actual loss value for better visualization.

| Base LLM | Res. | S-1 | Avg | GQA | POPE | MMB | MMS | MMVP |
|---|---|---|---|---|---|---|---|---|
| Llama-3-8B | 336 | ✗ | 55.5 | 56.6 | 84.0 | 64.3 | 46.0 | 26.7 |
| Llama-3-8B | 336 | ✔ | 60.5 | 60.7 | 84.5 | 72.9 | 51.3 | 33.3 |

Table 5: Ablation for the stage one (S-1). Both models are trained using 4M instruction tuning data. The model with the pre-training stage shows faster convergence and superior performance.

ception, where LLaVA-1.5-7B fails to recognize the color of small objects and the number of objects outside the image center. For instance, LLaVA-1.5-7B incorrectly identified the color of the NBA player's socks. The second row shows examples of logical reasoning, where the lack of fine-grained perception ability leads LLaVA-1.5-7B to failure in tasks that rely on it, such as edge object perception and reasoning, and highlighting regions in images. In contrast, our HaploVL, fusing raw image embeddings after the patch embedding layer, enhances its ability to perceive fine-grained image information. Therefore, it shows better performance on tasks relying on the capability of fine-grained perception.

**Is it possible to use the next token prediction loss directly?** To validate the effectiveness of the modal expansion, we directly optimized the model using next-token prediction loss without the first stage. As shown in Figure 6, the model converged slowly when optimized directly, as it had to perform both modality fusion and text generation simultaneously. In contrast, the model with the modal expansion stage converged significantly faster. Furthermore, as shown in Table 5, we found that the model without the modal expansion stage exhibits a 4.3% performance drop.

### 4.4. Visualization study

In order to investigate whether text embeddings can dynamically capture visual clues, we visualize the attention map between text embeddings and visual embeddings after the pre-decoder, as illustrated in Figure 5. It is observable that the text exhibits an automatic response to regions of higher relevance. For example, it demonstrates responsiveness to objects located at the image edges as well as to textual el-

ements within the image. These findings suggest that the early fusion mechanism of our single-transformer model is effective for fine-grained perception tasks, thereby corroborating the results presented in Table 4.

## 5. Conclusion and Discussion

This work presents a simple baseline for the multi-modal model with a single transformer architecture and a corresponding efficient training approach. By fusing raw vision and text embeddings in the early stages, our model enhances its fine-grained perception capabilities, enabling it to capture subtle relationships in the image better. Furthermore, our model builds upon prior knowledge from pre-trained single-modal models. This allows it to achieve superior performance with relatively few training tokens and bridge the performance gap between single-transformer multi-modal models and compositional models. Consequently, it demonstrates the potential of single-transformer architectures for multi-modal tasks.

We anticipate that future work can further improve the performance of our model by leveraging more data, setting longer context lengths, and adopting dynamic resolution as representative studies (Wang et al., 2024a). In addition, future work can employ such unified multi-modal models to act as agents (Yang et al., 2024b) for tool use. Furthermore, our qualitative results (Table 10 and Table 11) reveal that models trained on the same 665K instruction data exhibit similar response templates. This suggests that the patterns present in the data can impact the model's behavior. Therefore, future research can focus on developing more diverse formats to enhance the flexibility of LMMs. Plus, the alignment technique (Ouyang et al., 2022; Liao et al., 2024; Rafailov et al., 2024) can be used to align the LMMs with human preference. Finally, we envision that our model can be extended to generation tasks using a similar approach, thereby achieving a unified understanding and generation framework without the need for separate encoders (Radford et al., 2021; Zhai et al., 2023) and decoders (Podell et al., 2023) in a training efficient way.

## Acknowledgments

This work is supported by the National Natural Science Foundation of China (No. 62422606, 62201484).

## Impact Statement

This paper presents work whose goal is to advance the field of Machine Learning. There are many potential societal consequences of our work, none which we feel must be specifically highlighted here.

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

In the appendix, we first provide more details in Appendix A and more results in Appendix B. Secondly, implementation details are presented in Appendix C. Finally, more qualitative results are showcased in Appendix D.

## A. Model Details

**Current token prediction loss.** The PyTorch-like pseudo code for the proposed current token prediction loss is presented in Algorithm 1. The target for a text embedding is derived from the current token index instead of the next token index, which is a key distinction from the next token prediction loss (Radford et al., 2019).

---

**Algorithm 1** Current Token Prediction Loss

```
def loss(hidden_state, target_ids, embed_tokens):
    """
    hidden_state: [N, C]
    target_ids: [N]
    embed_tokens: [K, C]
    # N: the sequence length
    # K: the vocabulary size
    # C: the dimension of hidden_state
    """
    # get logits
    logits = hidden_state @ embed_tokens.transpose(-2, -1) # [N, K]
    # scale logits
    logits = logit_scale * logits
    # calculate loss
    loss = F.cross_entropy(logits, target_ids)
    return loss
```

---

embed_tokens is frozen.

**The positional embedding.** Positional embedding plays a crucial role in models (Dosovitskiy et al., 2021; Yang et al., 2022; Zhou et al., 2024) based on the attention mechanism (Vaswani et al., 2017), as it enables the self-attention module to capture the spatial relationship between embeddings. Our pre-decoder adopts a similar architecture to CLIP-ViT-L (Radford et al., 2021) but with the extra capability of accepting both image and text inputs. Therefore, pre-decoder must consider text position information in addition to image position information. To address this, we retain the learnable position embedding for image embeddings in the patch embedding layer and incorporate Rotary Position Embedding (RoPE) (Su et al., 2024) in each self-attention layer to inject positional information into the multi-modal embeddings.

## B. More Results

**Data in the pre-training stage.** The pre-training stage is a critical step in our training receipt since it enables the pre-decoder to acquire new text knowledge while retaining its inherited vision knowledge. The data composition in this stage can impact the model in processing multi-modal inputs, as it influences the model to learn effective representations of text and images. Therefore, we conducted an exploratory analysis of the training data used in this stage. Specifically, we omitted the alignment stage and proceeded directly to instruction tuning after completing the pre-training stage. Initially, we employed Vicuna-7B (Chiang et al., 2023) as the language model and CLIP-ViT-L-14 (Radford et al., 2021) with an input resolution of 224 as the vision teacher. We directly mix 665 K instruction data and 558 K pre-train data of LLaVA-1.5, termed 'mix-v1'. When training with this data, the average result was 59.0%. In this setup, the image position was fixed in the sequence, which may lead the model to learn shortcuts related to the position. This can hinder multi-modal fusion since the model may not learn to effectively integrate image and text embeddings. Therefore, we created an interleaved sequence by randomly combining 665 K instruction data and 558 K pre-train data, denoted as 'mix-v2', where each sequence may have multiple images. This data resulted in an average performance of 59.2% across multiple benchmarks. To further investigate the effectiveness of interleaved data in improving the performance on subsequent understanding tasks, we collected 730 K samples from MMC4 (Zhu et al., 2023b) for training. Our findings indicate that using interleaved data for modal expansion improves the performance on downstream tasks, particularly on ScienceQA (Lu et al., 2022b).

Next, we expanded the input resolution and used CLIP-ViT-L-14 (Radford et al., 2021) with an input resolution of 336 as the vision teacher for pre-training. When training with the 'mix-v2' data, the increased resolution resulted in a 1.5% improvement in average performance, with notable gains on GQA (Hudson & Manning, 2019), MMBench (Liu et al., 2024d), and MMStar (Chen et al., 2024a). At this resolution, we found that replacing MMC4 resulted in a similar improvement to that observed at the 224 resolution. Furthermore, we combined 730 K data from MMC4, 665 K instruction data, and 558 K

| LLM | Res. | Data-S1 | Avg | GQA | SQA | POPE | MMB | MMS |
|---|---|---|---|---|---|---|---|---|
| Vicuna-7B | 224 | mix-v1 | 59.0 | 59.0 | 61.7 | 84.6 | 56.9 | 32.8 |
| Vicuna-7B | 224 | mix-v2 | 59.2 | 60.7 | 63.7 | 83.6 | 55.2 | 32.8 |
| Vicuna-7B | 224 | mmc4 | 59.4 | 59.1 | 65.5 | 82.9 | 57.3 | 32.5 |
| Vicuna-7B | 336 | mix-v2 | 60.7 | 61.8 | 63.8 | 84.6 | 59.5 | 33.6 |
| Vicuna-7B | 336 | mmc4 | 61.1 | 61.9 | 67.0 | 84.4 | 58.3 | 33.7 |
| Vicuna-7B | 336 | mix-v3 | 61.2 | 61.2 | 67.5 | 84.8 | 59.0 | 33.6 |
| Vicuna-7B | 336 | mix-v4 | 61.3 | 62.0 | 67.3 | 84.7 | 59.5 | 33.3 |
| Llama-3-8B | 336 | mix-v4 | 64.2 | 63.1 | 70.5 | 84.8 | 63.0 | 39.4 |

Table 6: Ablation for data used in the first stage (Data-S1). The quantity of instruction data used in the second stage is 665 K. 'Res.' denotes the resolution of the vision teacher.

pre-train data to create 'mix-v3'. This mixture of data only brings a marginal improvement. Because our goal is to teach the model new text knowledge, we deemed it necessary to incorporate pure text data into the training process. Therefore, we mixed 665 K instruction data, 558 K pre-train data, and 600 K pure text data (Computations, 2023; Taori et al., 2023; ShareGPT, 2023) to create 'mix-v4'. Training with this data resulted in a slight performance improvement. Nevertheless, we confirm the importance of using pure text data to teach the model new knowledge. Building on this data, we replaced Vicuna-7B with Llama-3-8B (Dubey et al., 2024) as the LLM, which brings a significant improvement in performance from 61.3% to 64.2%. This suggests that the ability conditioned on multi-modal sequences for effective reasoning is crucial in multi-modal understanding and reasoning.

**Zero-shot accuracy on ImageNet.** In the pre-training stage, we leverage the vision encoder of CLIP-ViT-L (Radford et al., 2021) as a teacher model to teach the visual knowledge for pre-decoder. To evaluate the effectiveness of this stage, we assess the zero-shot accuracy on ImageNet (Deng et al., 2009) in Table 7. Specifically, when optimized directly using the next token prediction loss, pre-decoder fails to retain image classification capabilities as shown in the first line. In contrast, with the first stage, pre-decoder exhibits a minimal performance drop compared to its teacher model. This outcome confirms the efficacy of the first stage in our training receipt.

| LLM | Res. | w/ S1 | Data-S1 | Avg | IN1K | IN1K-Teacher |
|---|---|---|---|---|---|---|
| Llama-3-8B | 336 | × | - | - | 0.1 | 76.6 |
| Vicuna-7B | 224 | ✓ | mix-v2 | 59.2 | 71.7 | 75.5 |
| Vicuna-7B | 224 | ✓ | mmc4 | 59.4 | 71.6 | 75.5 |
| Vicuna-7B | 336 | ✓ | mix-v2 | 60.7 | 72.3 | 76.6 |
| Vicuna-7B | 336 | ✓ | mmc4 | 61.1 | 73.6 | 76.6 |
| Vicuna-7B | 336 | ✓ | mix-v3 | 61.2 | 73.6 | 76.6 |
| Vicuna-7B | 336 | ✓ | mix-v4 | 61.3 | 71.9 | 76.6 |
| Llama-3-8B | 336 | ✓ | mix-v4 | 64.2 | 72.4 | 76.6 |

Table 7: Zero-shot accuracy on ImageNet-1K (IN1K) (Deng et al., 2009) after the pre-training stage. 'w/ S1' denotes the model tuned by the pre-training stage. 'Avg' refers to the average results on GQA (Hudson & Manning, 2019), ScienceQA-IMG (SQA) (Lu et al., 2022b), POPE (Li et al., 2023c), MMBench (MMB) (Liu et al., 2024d), and MMStar (Chen et al., 2024a) benchmark. Vision teacher is the CLIP-ViT-L (Radford et al., 2021), and 'IN1K-Teacher' denotes its zero-shot accuracy on ImageNet-1K. The first line is the results of the model optimized directly by the next-token prediction loss.

## C. Implementation Details

### C.1. Experiment Settings

We summarize the training settings of each stage in Table 8 and Table 9. To maintain the training stability of the pre-training stage, we reduce the value of $\beta_2$ as Siglip (Zhai et al., 2023) rather than change the model structure as Chameleon (Team, 2024). For HaploVL-7B, we train it as LLaVA (Liu et al., 2024c). After pre-training, we first tune the connector between the pre-decoder and post-decoder using 558 K caption data (Liu et al., 2024a) and then fully tune the model using 665 K

instruction data (Liu et al., 2024a). For HaploVL-8B with the ability to input any resolution, we first tune the whole model using 1.2 M caption data (Chen et al., 2023) because the prior knowledge of the pre-decoder doesn't have any-resolution vision knowledge. Then, we tune the model using 4 M instruction data. For HaploVL-8B-MI that supports the multi-image and video input, we continue training the single-image model (HaploVL-8B) using the mix of interleaved data and single-image data.

| config | value |
|---|---|
| optimizer | AdamW (Loshchilov & Hutter, 2019) |
| base learning rate | $1.0 \times 10^{-4}$ |
| weight decay | $1.0 \times 10^{-4}$ |
| optimizer momentum | $\beta_1 = 0.9, \beta_2 = 0.95$ |
| batch size | 256 |
| learning rate schedule | cosine decay (Loshchilov & Hutter, 2017) |

Table 8: The first stage setting.

| config | value |
|---|---|
| optimizer | AdamW (Loshchilov & Hutter, 2019) |
| base learning rate | $2.0 \times 10^{-5}$ |
| weight decay | 0.0 |
| optimizer momentum | $\beta_1 = 0.9, \beta_2 = 0.98$ |
| batch size | 128 |
| learning rate schedule | cosine decay (Loshchilov & Hutter, 2017) |

Table 9: The second stage setting.

## D. Qualitative Results

We provide additional examples that demonstrate the capabilities of HaploVL in various tasks, including visual fine-grained perception, logistic reasoning, and image captioning. As shown in Table 10, HaploVL excels in fine-grained visual perception, including recognizing subtle color differences, object locations, and object quantities. Furthermore, HaploVL demonstrates correct logistic reasoning based on visual information, as illustrated in Table 11. This is attributed to its exceptional fine-grained perception ability, which is facilitated by early fusion. In terms of image captioning, HaploVL generally produces accurate descriptions, as shown in Table 12. Notably, HaploVL-8B pays closer attention to image details compared to HaploVL-7B, due to the higher quality and quantity of its instruction data. Interestingly, models trained on 665 K instructions from LLaVA (Liu et al., 2024a) exhibit a similar response pattern, suggesting that the inherent patterns in the data can influence the behaviors of the model.

**Visual fine-grained perception examples:**

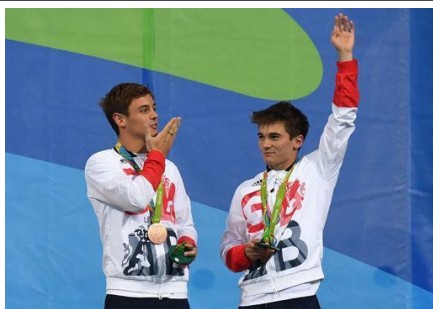

| | |
|---|---|
| User | What color is the ribbon that the man on the right is holding? |

| | |
|---|---|
| **LLaVA** | The man on the right is holding a **gold** ribbon. (✗) |
| **Chameleon** | The man on the right is holding a **white and red** ribbon. (✗) |
| **EVE** | The ribbon that the man on the right is holding is **green**. (✔) |
| **HaploVL-7B** | The man on the right is holding a **green** ribbon. (✔) |

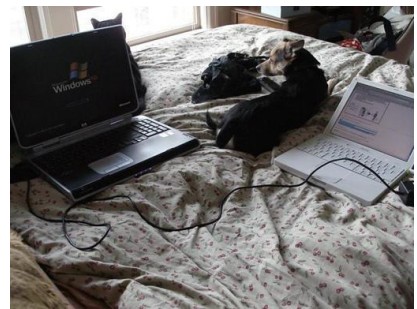

| | |
|---|---|
| User | Where is the cat? Options: top-right, top-left, bottom-left, bottom-right |

| | |
|---|---|
| **LLaVA** | Top-left. (✔) |
| **Chameleon** | The cat is on top of the blanket.(✗) |
| **EVE** | Top-right. (✗) |
| **HaploVL-7B** | Top-left. (✔) |

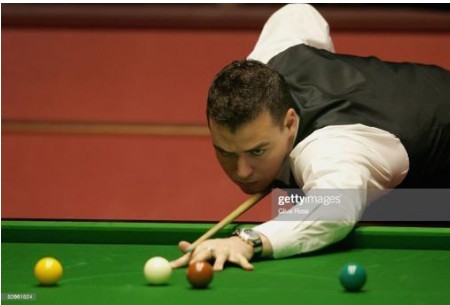

| | |
|---|---|
| User | How many balls are on the table? Options: A: 6, B: 4, C: 5, D: 3 |

| | |
|---|---|
| **LLaVA** | A. (✗) |
| **Chameleon** | The answer is B: 4. There are four balls on the table.(✔) |
| **EVE** | A. (✗) |
| **HaploVL-7B** | B. (✔) |

Table 10: Visual fine-grained perception examples. HaploVL excels in recognizing subtle colors, object locations, and object quantities.

**Visual logistic reasoning examples:**

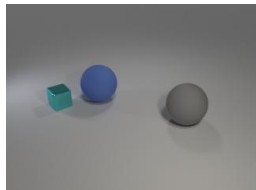

| | |
|---|---|
| User | Subtract all large gray things. Subtract all small brown metallic balls. How many objects are left? |

| | |
|---|---|
| **LLaVA** | After subtracting all large gray objects and small brown metallic balls, there are three objects left on the table: two blue balls and one gray ball. (✗) |
| **Chameleon** | There are 12 objects left. (✗) |
| **EVE** | There are three objects left: a blue sphere, a green cube, and a gray sphere. (✗) |
| **HaploVL-7B** | After subtracting all large gray things and small brown metallic balls, there are two objects left: a blue sphere and a green cube. (✔) |

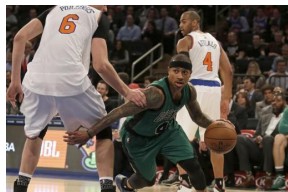

| | |
|---|---|
| User | What is the basketball player on the left doing in the image? |

| | |
|---|---|
| **LLaVA** | The basketball player on the left is trying to steal the ball from the player on the right. (✔) |
| **Chameleon** | The basketball player on the left is dribbling the ball with his left hand and is about to shoot the ball with his right hand. (✗) |
| **EVE** | The basketball player on the left is holding the ball and dribbling it while standing on the court. (✗) |
| **HaploVL-7B** | The basketball player on the left is attempting to steal the basketball from the player on the right. (✔) |

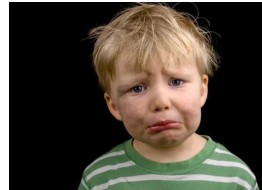

| | |
|---|---|
| User | What will happen next? |

| | |
|---|---|
| **LLaVA** | In the image, a young boy with a green shirt is crying and **has his eyes closed.** It is not possible to predict the exact next event, but it is likely that the boy's emotional state will continue, and he may eventually open his eyes and explain the reason for his **crying**. Alternatively, he might be comforted by someone or something, which could help him stop crying. The situation highlights the importance of empathy and support for children during their emotional moments. (✔, Hallucinations) |
| **Chameleon** | The image shows a young boy with **his mouth wide open**. He has light brown hair and is wearing a green striped t-shirt. There is a black background. He is likely to **blow out some candles on a cake.** (✗, Hallucinations) |
| **EVE** | In the image, a young boy with blond hair is **making a funny face**, possibly sticking out his tongue. It is not possible to predict exactly what will happen next, but it is likely that the boy will continue to make a funny face or gesture, possibly to entertain himself or others around him. The boy's facial expression suggests that he is in a playful mood, and he might engage in some lighthearted antics or jokes. (✗) |
| **HaploVL-7B** | In the image, a young boy with blonde hair is **shown crying with his eyes wide open**. It is difficult to predict exactly what will happen next, but considering the context of the image, it is likely that the boy's emotions will continue to be expressed through his tears. He might be expressing his feelings about something that has upset him or causing him distress. Alternatively, the boy could be reacting to a recent event or situation that has affected him deeply. The image captures **a moment of vulnerability and emotionality**, and it is important to be empathetic and supportive when encountering such situations. (✔) |

Table 11: Visual logistic reasoning examples. HaploVL shows correct reasoning for the open-ended questions.

**Visual caption example:**

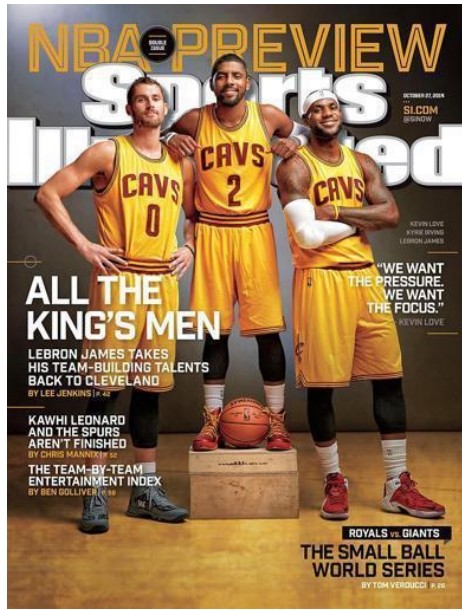

| User | Describe this image in detail. |
| --- | --- |
| **LLaVA** | The image features three professional basketball players posing for a photo on the cover of an **NRA** Sports magazine. They are all wearing yellow sports jerseys, showcasing their team spirit. The players are standing close to each other, with one player on the left, another in the middle, and the third on the right side of the image. 

 In the background, there is a sports ball, likely a basketball, placed near the center of the scene. The players are all smiling and appear to be enjoying the moment, capturing their camaraderie and dedication to the sport. (Distracted by text in images) |
| **Chameleon** | This image features two posters advertising the ”All-Star Weekend” of the National Basketball Association (NBA). The image is a portrait orientation and features two advertisements in the center of the image. 

 The advertisement on the left is titled ”KINGS” in white text with a bold, black outline. **The text is placed above a black-and-gold basketball that is shown from the side, with the black side facing the viewer. The advertisement below the title features two athletes, one with their back to the viewer and one facing forward. The athlete on the left is wearing a yellow Nike jersey with black sleeves and a black number ”20” on the back. They are holding a black basketball in their right hand and have their left arm raised in the air. The athlete on the right is wearing a gold Nike jersey with black trim and a black number ”35” on the back. They are also holding a black basketball in their right hand and have their left arm raised in the air. The athletes are both wearing black shorts and white socks. The background of the advertisement is a black wall with a large, yellow text that reads ”ALL-STAR WEEKEND” in bold, white text.** 

 Both advertisements feature the Nike logo in the top left corner of the advertisement, and the text ”Nike” is printed in white in the top right corner of each advertisement. (Hallucinations) |
| **EVE** | The image features three men dressed in yellow basketball uniforms, standing close together and posing for a picture. They are all wearing shorts and sports shoes, and one of them is holding a basketball. The men appear to be athletes, likely part of a basketball team. 

 The image also includes a few other elements, such as a sports ball located near the center of the scene, and **a couple of books placed on the ground. The books are positioned on the left side of the image, and they seem to be unrelated to the main focus of the scene,** which is the three men in their basketball uniforms. (Hallucinations) |
| **HaploVL-7B** | The image is a magazine cover featuring three basketball players from the Cavaliers team. The cover is titled “Sports Illustrated” and has a subtitle “All the King’s Men” with a tagline ”Lebron James takes his team-building talents to Cleveland.” The three players are standing next to each other, with the player in the middle holding a basketball. The cover also includes a note that the Cavaliers have won the NBA championship and a mention of the team-by-team entertainment index. The magazine cover is designed to attract readers interested in basketball and the Cavaliers team. (More consistent with the image) |

Table 12: Visual caption example. HaploVL provides a description that is more consistent with the image.

**Visual question answering example:**

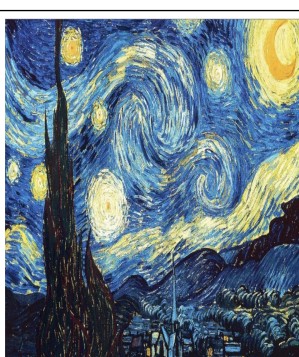 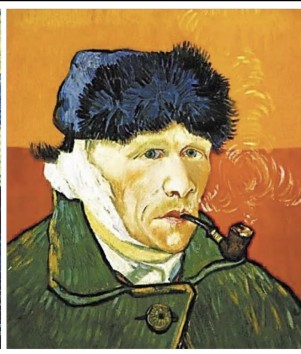 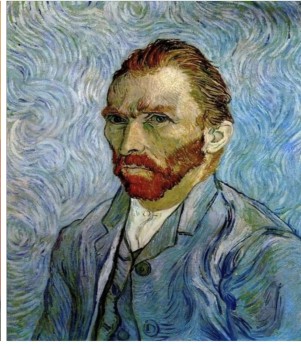

| User | Do you know who drew these paintings? Please explain each image. |
|------|------|
| **HaploVL-8B-MI** | The first painting is a famous work by Vincent van Gogh, titled "The Starry Night". It depicts a swirling night sky with bright stars and a bright crescent moon, set against a backdrop of dark, rolling hills and a small village with lit windows. The painting is characterized by its bold and expressive brushstrokes, and vibrant use of color. It is widely regarded as one of the most iconic and influential works of post-impressionist art. The second painting is also by Vincent van Gogh, titled "Self-Portrait with Pipe". It shows a man with a beard and pipe, wearing a blue hat, a white collar, and a green coat. The background is a solid orange color, which creates a strong contrast with the man's features. The painting is characterized by its bold and expressive use of color, as well as its swirling brushstrokes, which give it a dynamic and vibrant quality. The third painting is also by Vincent van Gogh, titled "Self-Portrait". It shows a man with a beard and pipe, wearing a blue suit and a white shirt. The background is a swirling pattern of blues and greens, which gives the impression of a dynamic and vibrant energy. The painting is characterized by its bold and expressive use of color, as well as its swirling brushstrokes, which give it a dynamic and vibrant quality. In summary, the first painting is "The Starry Night" by Vincent van Gogh, which depicts a swirling night sky with bright stars and a bright crescent moon. The second painting is "Self-Portrait with Pipe" by Vincent van Gogh, which shows a man with a beard and pipe, wearing a blue hat, a white collar, and a green coat. The third painting is "Self-Portrait" by Vincent van Gogh, which shows a man with a beard and pipe, wearing a blue suit and a white shirt, set against a swirling pattern of blues and greens. |

Table 13: Visual question answering example. HaploVL able to process multiple images.

