# OpenReview forum: "HaploVL: A Single-Transformer Baseline for Multi-Modal Understanding"
_ICML.cc/2025/Conference — ICML 2025 poster_

### Official Review · Reviewer_TtK1 · 2025-03-11

**Overall Recommendation:** 3

**Summary:**

- This paper introduces HaploVL, a multimodal model with a dual Transformer decoder for joint vision-language processing.
- The proposed two-stage training distills knowledge from a pre-trained model into the first decoder, integrating text and vision.
- HaploVL extends the EVE approach by reusing aligned vision tokens in a second decoder module.
- Built on CLIP-ViT-L and Llama-3-8B, HaploVL outperforms baselines across multiple multimodal benchmarks.

**Claims And Evidence:**

- The first main claim in lines 24–27 and 96–99 is overclaimed. The authors state, “First, we propose a new early-fusion LMM that can fuse multi-modal inputs in the early stage and respond to visual instructions in an autoregressive manner.” However, this early-fusion design is conceptually similar to related work such as EVE, and although this work employs a different teacher encoder to incorporate prior knowledge and reuse aligned tokens in a second decoder, the core idea of the first decoder remains the same.
- The second main claim is also inaccurate. The claim in lines 21–24 that the model is an “end-to-end large multi-modal model in a single transformer” is misleading. In reality, the proposed system is a dual-decoder architecture trained in two stages with different objective functions at each stage. In a truly end-to-end system, all components would be jointly trained from the start with a unified objective, and calling the dual decoder system a “single transformer” is imprecise.
- Additionally, the authors argue that the method is encoder-free. However, they do make use of pre-trained encoder parameters. In specific, l.188-189 ‘For the input text $Xt$, we leverage the pre-trained LLM’s embedding matrix W to convert each text token into a vector within LLM’s space $R^l$.’, and l.207-210 ‘Notably, although the pre-decoder inherits prior knowledge from a vision encoder, it differs from the vision encoder.’ The subsequent explanation in lines 210–215 does not sufficiently justify the claim that the architecture is encoder-free.

**Essential References Not Discussed:**

[1] Yao, Y., Yu, T., Zhang, A., Wang, C., Cui, J., Zhu, H., Cai, T., Li, H., Zhao, W., He, Z. and Chen, Q., 2024. Minicpm-v: A gpt-4v level mllm on your phone. arXiv preprint arXiv:2408.01800.

**Experimental Designs Or Analyses:**

- An ablation study is needed using a single decoder with a size equal to the combined size of the two decoder layers—that is, an EVE-style setup where the decoder's capacity matches that of the two decoders used in this work.
- An ablation study is also required for an encoder–decoder–based approach that utilizes the same pre-trained encoders employed as teachers in this work (e.g., CLIP for vision and Llama-3 for text).
- In Table 4 and lines 358–359, the experiment “to verify whether the LMM using one single transformer has advantages over separate models” is important, but it lacks setup details. Specifically, clarify whether the single transformer in EVE-7B has the same number of decoder layers and/or parameters as the combined decoders in HaploVL-7B, rather than just matching the second decoder (the LLM).

**Methods And Evaluation Criteria:**

Given the proposed method use Llama-3-8B as base LLM, in Table 2 there should be Llama-3-8B based VLMs, e.g. MiniCPM-V2.5 [1] and Llama-3.2.

**Other Comments Or Suggestions:**

l.185-186: ‘it needs to acquire prior textual knowledge the Llama model’ → ‘it needs to acquire prior textual knowledge from the Llama model’

**Other Strengths And Weaknesses:**

- The presentation and experiments are sufficient.
- The technique’s novelty is modest, offering only minor innovations over EVE when combined with a customized training recipe.
- The writing—particularly the claims about key contributions—requires more careful revision. There are numerous overclaims throughout the manuscript, the authors should ensure that their claims accurately reflect their contributions.

**Questions For Authors:**

See above.

**Relation To Broader Scientific Literature:**

See below the ‘Essential References Noe Discussed’ section

**Theoretical Claims:**

- Many arguments in this work are speculations and lack theoretical or empirical support.
    - l.47-l.50 ‘Our model fuses the vision and text embeddings at an early stage, enabling text embeddings to **autonomously acquire the necessary vision cues**.’
    - l.117-120 ‘our HaploVL fused the visual and textual input in the early stage and **extracts the necessary vision information based on the text input.**’
    - l.244-547 ‘When the text and image are jointly input into the pre-decoder in a mixed way, **semantic text embeddings can autonomously acquire the necessary vision cues from raw vision embeddings**.’
- The teacher model of the text embedding from pre-deocder seems not introduced in l.248-l.274.

---

> ### Author Rebuttal · Authors · 2025-04-01
>
> We appreciate your suggestions and feedback. In the following, we respond to the major concerns.
>
> * Q1: This early-fusion design is conceptually similar to related work such as EVE, offering only minor innovations over EVE when combined with a customized training recipe.
>
> **Response:** We distinguish our HaploVL from EVE through the following differences:
>
> | Difference      | EVE                                                           | Ours                                                                                        |
> |-----------------|---------------------------------------------------------------|---------------------------------------------------------------------------------------------|
> | Architecture    | Uses multiple attention layers to tokenize images (see Figure 3 in EVE's paper) | Utilizes a simple MLP to tokenize images, ensuring simplicity in inference                  |
> | Methodology     | Does not inherit vision knowledge, requiring 35M data for training                | Inherits vision knowledge, needing only 1.2M samples to achieve performance surpassing EVE |
> | Performance     | Shows a significant performance gap with separate models, e.g., 28.2 on MMStar                         | Achieves performance comparable to other separate models, e.g., 34.5 on MMStar                                    |
>
> These differences highlight our architectural innovations and efficiency in training, which contribute to the superior performance of HaploVL compared to EVE.
>
> * Q2: The claim about "end-to-end large multi-modal model in a single transformer"
>
> **Response:** We acknowledge that the model employs a dual-decoder architecture during training. However, this dual-decoder setup is used exclusively for the training phase. **During inference, both the image and text inputs pass through the same single transformer model. This aligns our approach more closely with end-to-end processing.** Other models like EVE also utilize multi-stage training strategies.
>
> Furthermore, we have experimented with jointly training all components from the start using a unified objective. The results indicate that **"training all components from the start using a unified objective" yields lower performance compared to our two-stage training approach.** Specifically, the average score across 5 benchmarks was 60.5 for the two-stage method, versus 55.5 for the one-stage joint training.
>
> * Q3: The subsequent explanation in lines 210–215 does not sufficiently justify the claim that the architecture is encoder-free.
>
> **Response:** First, we do not claim that HaploVL is encoder-free. Second, we outline the differences between our pre-decoder and Vision Transformer (ViT) architecture:
>
> | Aspect                | ViT                          | Our Pre-decoder                                |
> |-----------------------|------------------------------|------------------------------------------------|
> | Input                 | Image-only                   | Image and text                                 |
> | Positional embeddings | Learnable positional embeddings | Rotary positional embeddings (RoPE)             |
> | Attention block       | Differs from LLM attention block | Same attention block as post-decoder           |
> | Performance           | Shows inductive bias, low fine-grained perception | Better fine-grained perception due to early fusion |
>
> * Q4: There should be Llama-3-8B based VLMs, e.g. MiniCPM-V2.5 and Llama-3.2.
>
> **Response:** We will include the results of MiniCPM-V2.5 and Llama-3.2 in our main table for comparison. However, it should be noted that MiniCPM-V2.5 utilizes 778M training data, whereas our best model is trained on less than 6M samples. Additionally, **we fully leverage open-source data, which ensures our model's ease of reproducibility.**
>
> * Q5: The teacher model of the text embedding.
>
> **Response:** It is the text embeddings of the language model. We will clarify this in the final version.
>
> * Q6: Ablation study for using the same pre-trained encoders and LLM.
>
> **Response:** First, we report the total parameters of the two decoders. The parameters of our two decoders are almost equivalent to those of EVE (7.3B vs 7B).
> Second, we conducted comparisons using the same LLM (Vicuna-7B) and equivalent data with LLaVA-1.5-7B. Our findings demonstrate that HaploVL-Vicuna-7B exhibits superior performance compared to EVE-7B and surpasses LLaVA-1.5-7B on fine-grained perception tasks.
>
> | Model        | LLM       | SEED | MMStar | MMVP |
> |--------------|-----------|------|--------|------|
> | EVE-7B       | Vicuna-7B | 54.3 | 28.2   | 19.3 |
> | LLaVA1.5–7B  | Vicuna-7B | 66.1 | 30.2   | 21.3 |
> | HaploVL-7B   | Vicuna-7B | 67.5 | 34.5   | 24.7 |
>
> These ablation results validate our architectural innovations and demonstrate significant performance enhancements.

---

### Official Review · Reviewer_y5EY · 2025-03-12

**Overall Recommendation:** 4

**Summary:**

HaploVL is a large, single-transformer multi-modal model designed to overcome the limitations of existing models by integrating visual and textual inputs early on for efficient multi-modal comprehension.

They introduce an innovative pre-decoder model that merges visual patches with text embeddings at the initial stage.

Their two-stage approach utilizes knowledge distillation to maintain vision capabilities while fine-tuning with visual instruction data.

They outperform previous single-transformer models in their results.

**Claims And Evidence:**

This paper provides detailed experiment results and these results are convincing.

**Essential References Not Discussed:**

No.

**Experimental Designs Or Analyses:**

Part of the experimental design is reasonable.

1. The author proposes using a unique mask strategy and does not verify the effectiveness of this design.

2. The author needs to verify the effectiveness of the distillation module, particularly when initializing the pre-decoder with a well-trained vit.

**Methods And Evaluation Criteria:**

The authors propose a new model design using a single transformer and claim its practical significance.

**Other Comments Or Suggestions:**

No.

**Other Strengths And Weaknesses:**

1. The comparison method is not detailed enough and lacks performance comparison with mainstream methods.

2. Is this design necessary? This model still includes a vit and a large language model and an even more complex projector. Even compared to the early LLaVA 1.5, the performance improvement of the model is not very significant.


3. Can the effectiveness be validated on a smaller model? The strong language model may narrow the performance gap.

**Questions For Authors:**

No.

**Relation To Broader Scientific Literature:**

Earlier models like LLaVA and BLIP2 use separate vision encoders and language models. Subsequently, EVE and FUYU propose using only a single transformer to process multimodal input simultaneously.

This work also proposes a single transformer model design which consists of a pre-decoder and post-decoder.

**Theoretical Claims:**

There are no proofs in this paper that should be checked for correctness.

---

> ### Author Rebuttal · Authors · 2025-04-01
>
> Thank you for your suggestions and feedback. We respond to the major concerns in the following.
>
> * Q1: The comparison method is not detailed enough.
>
> **Response:** Due to page limitations, we have compared our model against several widely-recognized methods, which include very mainstream approaches. For single-transformer LMMs, we have also included comparisons with the latest models.
> To provide a more comprehensive comparison, we will add the results of more state-of-the-art models to our main table.
>
> However, regarding recent state-of-the-art separate LMMs, such as  QwenVL-2 and InternVL-2.5, it's important to note that these models used closed datasets, although their model weights were released. According to their technical reports, QwenVL-2 utilized 1.4 trillion tokens during pre-training, and InternVL-2.5-7B used 142 billion tokens.
> **In contrast, our HaploVL-7B relies solely on open-sourced data, using only 1.2 million samples (~7 billion tokens). This ensures that our model can be easily reproduced by the research community to build their single transformer models.**
>
> * Q2: The necessity of this design.
>
> **Response:**
> **Our design aims to explore a novel architecture that utilizes a single transformer during inference.** Building on the well-established vision knowledge acquired from web-scale image data by Vision Transformer, we leverage this by initializing our pre-decoder with ViT and employ it as the teacher model. This allows us to inherit its vision knowledge and significantly reduce the required data and training costs compared to other early-fusion and single-transformer models.
> For example, while single-transformer LMMs like EVE and Emu3 have notable performance gaps when compared to separated LMMs such as LLaVA, **our approach strives to narrow this performance gap**. By integrating the strengths of ViT in the initial stages, we can achieve enhanced efficiency and performance, making our model a compelling candidate despite the apparent complexity.
>
> * Q3: The strong language model may narrow the performance gap.
>
> **Response:**
> We conducted comparisons using the same LLM (Vicuna-7B) and equivalent data with LLaVA-1.5-7B. This can eliminate the effect of the strong language model. Our findings demonstrate that HaploVL-Vicuna-7B exhibits superior performance compared to EVE-7B and surpasses LLaVA-1.5-7B on fine-grained perception tasks.
>
> | Model        | LLM       | SEED | MMStar | MMVP |
> |--------------|-----------|------|--------|------|
> | EVE-7B       | Vicuna-7B | 54.3 | 28.2   | 19.3 |
> | LLaVA1.5–7B  | Vicuna-7B | 66.1 | 30.2   | 21.3 |
> | HaploVL-7B   | Vicuna-7B | 67.5 | 34.5   | 24.7 |
>
> **These results validate our architectural innovations rather than relying solely on the superior capabilities of a strong language model.** This demonstrates that our design contributes significantly to performance enhancement.

---

> > ### Comment · Reviewer_y5EY · 2025-04-02
> >
> > The rebuttal content provides clear contrasts, and the results indicate the effectiveness of its architecture, especially with the limited training data. Therefore, I think this work is worth accepting, and I will improve the rating.
> >
> > Therefore, I believe this work deserves approval and will improve my rating.

---

### Official Review · Reviewer_6jcN · 2025-03-13

**Overall Recommendation:** 4

**Summary:**

The paper introduces HaploVL, an early-fusion multi-modal model (LMM) that processes visual and textual inputs through a single-transformer architecture. Unlike traditional compositional LMMs that handle modalities separately, HaploVL integrates raw visual and textual embeddings at an early stage, leveraging a pre-decoder to extract visual cues from text-guided attention and a post-decoder for deeper multi-modal fusion. By inheriting prior knowledge from pre-trained vision and language models (e.g., CLIP-ViT and Llama-3), HaploVL achieves competitive performance on fine-grained perception and reasoning tasks (e.g., MMVP and MMStar benchmarks) while requiring significantly less training data and computational resources. The model outperforms existing single-transformer LMMs (e.g., Fuyu-8B, EVE-7B) and rivals compositional LMMs (e.g., LLaVA-1.5) in specific benchmarks.

Strengthens

- Native Multi-Modal Architecture: HaploVL eliminates the need for separate vision/text encoders (e.g., ViT + LLM pipelines), simplifying the design and reducing computational overhead.

- Efficient Early Fusion: By fusing visual and textual embeddings early, the model retains fine-grained visual details, enhancing performance on perception-heavy tasks (e.g., 4.9% improvement in fine-grained perception over LLaVA-1.5).
Data and Resource Efficiency: Leveraging pre-trained models significantly cuts training costs. For example, HaploVL-7B achieves superior results using only 1.2M training samples vs. EVE-7B’s 35M.

- Clear Methodology: The two-stage training (pre-training for vision-text alignment and fine-tuning for instruction following) is well-structured, and the ablation studies (e.g., resolution scaling, data mixtures) validate design choices.

**Claims And Evidence:**

Weakness：
- Limited Benchmark Comparisons:
Outdated Baselines: The paper focuses on older models (e.g., LLaVA-1.5, InstructBLIP) and lacks comparisons with recent state-of-the-art LMMs like InternVL2.5-7B or QwenVL2-7B, which achieve superior MMBench scores (>80).
Context-Length Limitations: The authors acknowledge that HaploVL underperforms LLaVA-OV due to restricted tokenization (2,304 vs. 7,290 tokens), suggesting scalability challenges.

- Unclear Impact of Training Strategy:
The decision to discard visual supervision in Stage 2 (post-decoder fine-tuning) lacks justification. Retaining visual losses might improve multi-modal alignment but risks overfitting.
Baseline Fairness:

- HaploVL-8B uses Llama-3-8B, while EVE-7B uses weaker Vicuna-7B. Performance gains could stem from Llama-3’s superior language capabilities rather than architectural innovations.

Questions：
1. Impact of Retaining Visual Supervision in Stage 2
 -If visual losses are retained during Stage 2, the model might better preserve fine-grained visual-text alignment. However, this could also:

- Improve Performance: By preventing catastrophic forgetting of visual features.
- Harm Performance: If textual instruction tuning dominates the loss, visual signals could introduce noise.

- The paper’s current approach (discarding visual losses) prioritizes language-focused instruction following. To resolve this, ablation experiments comparing both strategies are needed.

2. Missing Comparisons with Recent Models
The authors should include benchmarks against InternVL2.5-7B and QwenVL2-7B, which excel in MMBench (scores >80). HaploVL’s MMBench score of 75.0 (HaploVL-8B-MI) falls short, suggesting architectural or scalability limitations. Potential solutions:
- Expand tokenization capacity (e.g., longer context windows).
- Incorporate high-resolution training (beyond 672×672).

3. Fair Baseline Comparison
- To isolate the impact of the proposed architecture (vs. Llama-3’s superiority), the authors should: Re-train EVE-7B using Llama-3-8B under identical settings.
- Compare the revised EVE-7B (Llama-3) with HaploVL-8B. This ensures gains are attributable to HaploVL’s early fusion, not the LLM backbone.


4.Minor Issues
- Typo: Line 412: “Figure Figure 5.” → Correct to “Figure 5.”
- Clarity: Clarify resolution scales (e.g., 336 vs. 672 in Table 3) and tokenization limits in the main text.

**Essential References Not Discussed:**

NO

**Experimental Designs Or Analyses:**

SEE Claims And Evidence

**Methods And Evaluation Criteria:**

SEE Claims And Evidence

**Other Comments Or Suggestions:**

SEE Claims And Evidence

**Other Strengths And Weaknesses:**

SEE Claims And Evidence

**Questions For Authors:**

SEE Claims And Evidence

**Relation To Broader Scientific Literature:**

SEE Claims And Evidence

**Theoretical Claims:**

SEE Claims And Evidence

---

> ### Author Rebuttal · Authors · 2025-04-01
>
> Thank you for your insightful suggestions and feedback. We have responded to the key concerns in the details below.
>
> * Q1: Limited Benchmark Comparisons.
>
> **Response:** While recent state-of-the-art LMMs such as QwenVL2-7B and InternVL2.5-7B achieve higher MMBench scores, it is important to note that these models use closed data despite releasing their model weights. According to their technical reports, QwenVL2-7B utilized 1.4 trillion tokens during the pre-training phase, and InternVL2.5-7B used 142 billion tokens. In contrast, our HaploVL-7B model relies solely on open-sourced data, comprising only 1.2 million samples (~7 billion tokens). This significantly lower data usage underscores the efficiency of our approach. Moreover, **our reliance on open-sourced data ensures that our model can be easily reproduced by the research community to build their single transformer models.** Nevertheless, we acknowledge the importance of more recent benchmarks and will incorporate the results of these state-of-the-art models into our main table for comprehensive comparison.
>
> * Q2: Context-Length Limitations
>
> **Response:** The context length of HaploVL is indeed scalable. Specifically, with an input resolution of 336, the context length is 2048 tokens, and it extends to 6144 tokens when the maximum image size is 672x672. Even at a context length of 2048 tokens, HaploVL-Vicuna-7B (HaploVL-7B) demonstrates superior performance compared to LLaVA-1.5-7B on the MMVP, MMStar, and SEED benchmarks.
>
> | Method         | context-length | MMVP | MMStar | SEED |
> |----------------|----------------|------|--------|------|
> | LLaVA-1.5-7B   | 2048           | 21.3 | 30.3   | 66.1 |
> | HaploVL-7B     | 2048           | 24.7 | 34.5   | 67.5 |
>
> * Q3: Impact of Retaining Visual Supervision in Stage 2
>
> **Response:** We retain a vision loss by adding an image decoder during the second stage of our method. This approach is similar to the Masked Autoencoder (MAE), where image embeddings are decoded into RGB images. To assess the impact, we conducted experiments using the LLaVA-665K instruction data.
>
> | W/ visual loss | GQA  | MMStar |
> |----------------|------|--------|
> | True           | 60.8 | 34.0   |
> | False          | 62.5 | 34.5   |
>
> **Models trained with vision loss perform worse**, with GQA and MMStar scores dropping from 62.5 to 60.8 and from 34.5 to 34.0, respectively. This is because additional vision loss conflicts with the textual loss used in multimodal understanding. We will include this result in the latest version.
>
> * Q4: Baseline Fairness
>
> **Response:** Due to the high workload involved in reproducing the EVE model (35 million data samples and 2 A100-80G GPUs running for 9 days), **we compared the performance of HaploVL utilizing the same Vicuna-7B language model.** The results demonstrate that HaploVL-Vicuna-7B outperforms EVE-7B. Specifically, on the SEED benchmark, HaploVL-Vicuna-7B achieves 67.5%, while EVE-7B (based on Vicuna-7B) only scores 54.3%.
>
> Furthermore, when utilizing the same Qwen2.5-7B language model, our HaploVL-7B-Pro also surpasses the performance of the improved EVE-2.0 (Qwen2.5-7B).
>
> | Model         | LLM         | SEED  | POPE  |
> |---------------|-------------|-------|-------|
> | EVE-7B        | Vicuna-7B   | 54.3  | 83.6  |
> | HaploVL-7B    | Vicuna-7B   | 67.5  | 85.4  |
> | EVE-2.0-7B    | Qwen2.5-7B  | 71.4  | 87.6  |
> | HaploVL-7B-Pro| Qwen2.5-7B  | 75.0  | 88.7  |
>
> These results confirm that **the performance improvements are attributable to HaploVL’s architectural innovations rather than just the superiority of the LLM backbone.**

---

### Official Review · Reviewer_yfeb · 2025-03-18

**Overall Recommendation:** 3

**Summary:**

The paper proposes an early-fusion method for vision-language reasoning. They claim to have pre-decoder that extracts visual information from raw vision embeddings based on text input and a post-decoder to process fused multi-modal embeddings and generate text responses. The experiments suggest that the method is better than many state of the art methods, including EVE.

**Claims And Evidence:**

The experiments correctly reflect the claims of the paper.

**Essential References Not Discussed:**

None that I know of.

**Experimental Designs Or Analyses:**

The experiments look valid to the best of my knowledge.

**Methods And Evaluation Criteria:**

Yes, the proposed method seems reasonable and correct for the application.

**Other Comments Or Suggestions:**

I write the overall strength and weakness of the paper here

Positives

- The proposed method is simple and effective.
- The overall idea of early fusion is very interesting (though I am not very familiar with related work in this area and would seek help from other reviewers as well)
- The experiments show good gains compared to the competitive baseline

Negatives

- It is not clear why this method would take less data compared to EVE. Can the authors clarify more beyond stating that early fusion leads to efficient data usage.
- The limitations of the method are not clearly stated. Why can this method not be used for image generation if the setup is reversed, since the usage of visual and text modality looks symmetrical?

Overall, I like the paper and want it to be accepted. Please clarify on the negatives in the rebuttal phase.

**Other Strengths And Weaknesses:**

Please see questions for authors.

**Questions For Authors:**

Please see comments or suggestions.

**Relation To Broader Scientific Literature:**

Overall, vision-language learning is important for various applications. The proposed paper fits correctly in the overall literature.

**Theoretical Claims:**

No, I did not check theoretical claims. I believe the paper does not propose any theoretical claims either.

---

> ### Author Rebuttal · Authors · 2025-04-01
>
> Thank you for your valuable suggestions and feedback. We address the primary concerns as follows.
>
> * Q1: Clarify more beyond stating that early fusion leads to efficient data usage.
>
> **Response:** We propose utilizing a pre-decoder to fuse image and text data in the early stages of processing. **This pre-decoder is designed to inherit prior vision knowledge from a visual model (like ViT), thereby requiring minimal additional data.** The inherited prior vision knowledge enables the pre-decoder to effectively integrate visual information with textual data using fewer samples. Moreover, empirical results demonstrate that this method achieves superior performance with the same LLM and vision teacher, as evidenced by the MMStar benchmark scores (HaploVL-Vicuna-7B: 34.5 vs EVE-Vicuna-7B: 28.2).
>
> * Q2: Used for image generation.
>
> **Response:** Our current work verifies that this method is feasible for image understanding. In order to explore image generation, we introduce a vision loss by adding an image decoder during the second stage of our method. This approach is similar to the Masked Autoencoder (MAE), where image embeddings are decoded into RGB images. To assess the impact, we conducted experiments using the LLaVA-665K instruction data.
>
> | W/ visual loss | GQA  | MMStar |
> |----------------|------|--------|
> | True           | 60.8 | 34.0   |
> | False          | 62.5 | 34.5   |
>
> Models trained with vision loss perform worse, with GQA and MMStar scores dropping from 62.5 to 60.8 and from 34.5 to 34.0, respectively. While the symmetrical usage of visual and text modalities seems plausible, **the introduction of vision loss for image generation currently degrades the performance of our model in image understanding tasks. This is because additional vision loss conflicts with the textual loss used in multimodal understanding.** We plan future experiments to further explore and possibly mitigate these conflicts to successfully integrate image generation capabilities.

---

### Decision · Program_Chairs · 2025-05-01

**Decision:**

Accept (poster)

**Comment:**

This paper introduces HaploVL, an early-fusion multimodal LLM that processes visual and textual inputs through a single-transformer architecture. After rebuttal, it received scores of 3344, and one reviewer increased their score from 2 to 3.  Overall, the reviewers are happy about the paper after rebuttal, commenting that the overall idea of early fusion is interesting, and the experiments show good gains compared to competitive baselines.

On the other hand, I also share some concerns with the reviewers. That is, (1) the paper mainly compared with relatively old baselines, and results on more challenging benchmarks, such as those involving text-rich image understanding tasks that demand high-resolution image encoders, are not included to more comprehensively benchmark the model. (2) The writing—particularly the claims about key contributions—requires more careful revision. There are some overclaims in the manuscript that needs revision.

Overall, the AC thinks this paper still presents some interesting ideas, and the strengths outweigh the weaknesses, therefore, would like to recommend acceptance by the end.